# PIXEL-SPACE POST-TRAINING OF LATENT-DIFFUSION MODELS

## ABSTRACT

Latent diffusion models (LDMs) have made significant advancements in the field of image generation in recent years. One major advantage of LDMs is their ability to operate in a compressed latent space, allowing for more efficient training and deployment. However, despite these advantages, challenges with LDMs still remain. For example, it has been observed that LDMs often generate high-frequency details and complex compositions imperfectly. We hypothesize that one reason for these flaws is due to the fact that all pre- and post-training of LDMs are done in latent space, which is typically $8 \times 8$ lower spatial-resolution than the output images. To address this issue, we propose adding pixel-space supervision in the post-training process to better preserve high-frequency details. Experimentally, we show that adding a pixel-space objective significantly improves both supervised quality fine-tuning and preference-based post-training by a large margin on a state-of-the-art DiT transformer and U-Net diffusion models in both visual quality and visual flaw metrics, while maintaining the same text alignment quality.

## 1 INTRODUCTION

Diffusion models learn to sequentially denoise from random Gaussian noise to sharp images and have revolutionized the field of media generation and editing in recent years. Latent diffusion models represent the most popular type of diffusion model because of their efficiency and simplicity. State-of-the-art LDMs are typically pretrained on webscale data, resulting in "foundation models" (Rombach et al., 2022; Podell et al., 2023; Esser et al., 2024; Saharia et al., 2022; Imagen 3 Team, 2024; Dai et al., 2023; Ramesh et al., 2021; 2022; Betker et al., 2023).

These foundation models are then post-trained on a smaller, carefully curated dataset to improve quality through either supervised quality fine-tuning (SFT) (Dai et al., 2023) or human-in-the-loop preference modeling (Rafailov et al., 2024; Wallace et al., 2024; Meng et al., 2024). Post-training of image foundation models is also utilized to create new models for a variety of applications, including controllable generation (Zhang et al., 2023), editing (Sheynin et al., 2024), 3D generation (Poole et al., 2022), video generation (Singer et al., 2022; Girdhar et al., 2023), and many others.

To achieve efficiency and simplicity, LDMs use a pretrained variational autoencoder (VAE) to compress images into latent representations. For example, in the original LDM paper (Rombach et al., 2022), the authors used a conv-based VAE to compress a $512 \times 512 \times 3$ image to $64 \times 64 \times 4$. This representation significantly speeds up training and reduces computational cost as the denoising diffusion model now operates in the $64 \times 64 \times 4$ space instead of the original $512 \times 512 \times 3$ space ($48\times$ compression). However, this comes at the cost of lossy compression, which can result in inaccuracies in or loss of high-frequency details.

The research community has invested considerable effort in improving fine details, including scaling up the model, carefully curating fine-tuning datasets, increasing the latent channel dimension (Dai et al., 2023), and designing more powerful decoders (Betker et al., 2023).

In this paper, we take a step back and hypothesize that the artifacts in LDMs are partially caused by the fact that all pretraining, post-training, and inference steps are done on a lower-resolution latent space. With this assumption, we propose adding an additional supervision term in the original pixel space during post-training by decoding the latent representation back and combining it with the original latent loss term. This approach aims to improve the quality of generated images by

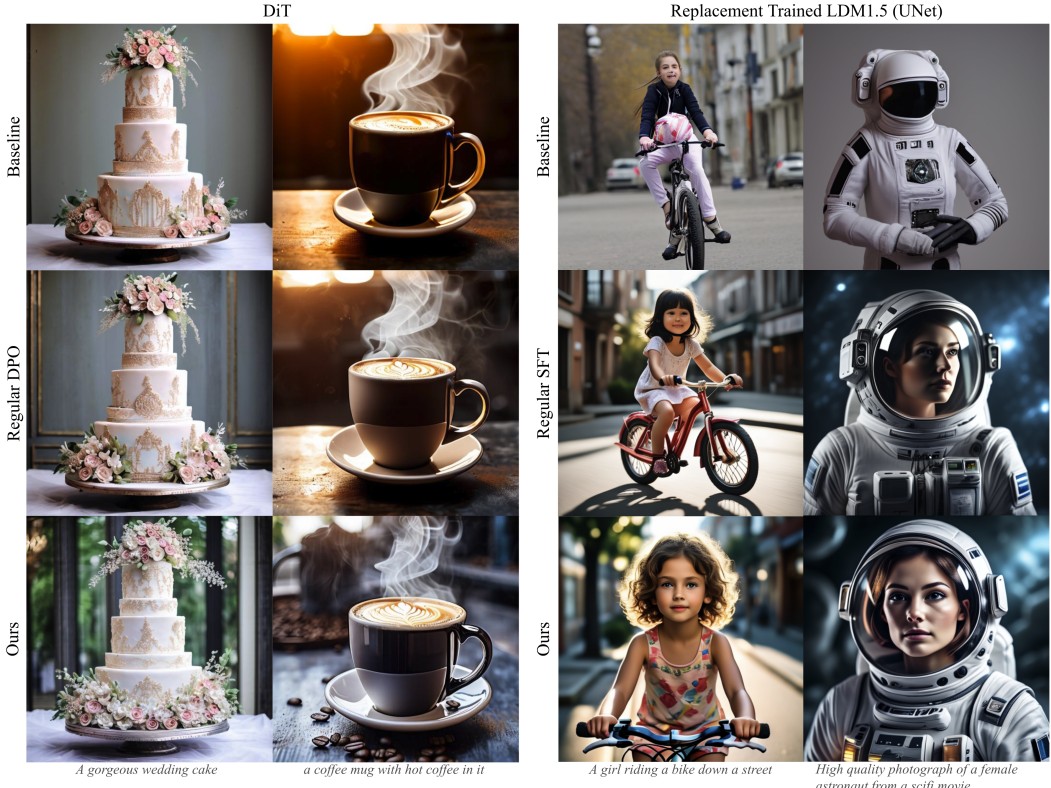

Figure 1: **Enhancing LDMs with pixel-space objectives.** We hypothesize that losses of details and artifacts in high-frequency details are partially caused by training on the lower-resolution latent space. We propose adding a pixel-space objective during LDM post-training. Our experiments show significant improvements in both DiT-based and UNet-based LDMs for reward-based and supervised fine-tuning.

providing additional guidance in the pixel space, which helps to mitigate the loss of high-frequency details and artifacts introduced by the compression of the latent space. Figure 1 demonstrates that our method can generate more stunning details when post-trained on the same dataset.

Through extensive experiments and independent human evaluation from annotators who have no knowledge of this project, we have found that the proposed method has the following advantages:

1. **Simplicity**: The proposed method does not modify the architecture of the diffusion denoising model and can be seamlessly integrated into any LDM-based model without introducing new parameters, making it flexible and efficient.

2. **Effectiveness**: Despite its simplicity, we found that the proposed method is surprisingly effective, resulting in a $18.2\%$ and $23.5\%$ improvements on visual appeal and visual flaws with supervised fine-tuning, and $17.8\%$ and $11.3\%$ improvements on preference-based fine-tuning on a DiT model on head-to-head A/B comparisons with the latent-space baseline.

3. **General applicability to models**: The proposed method performs remarkably well in both DiT and U-Net based LDMs (Rombach et al., 2022; Dai et al., 2023).

4. **General applicability to post-training methods**: The proposed method works well on both supervised fine-tuning and reward-based fine-tuning, and can be easily added to the future post-training methods researchers develop.

A secondary contribution of this paper is that we are the first paper that extends the recently proposed SimPO (Meng et al., 2024) preference optimization post-training technique to the image generation task and shows its effectiveness on the diffusion-based image generation domain.

## 2 RELATED WORK

A comprehensive review of diffusion models is out of the scope of this section. Interested readers are referred to Fuest et al. (2024) and Chan (2024). Here we highlight a few works that are closest-related to us.

### 2.1 TEXT-TO-IMAGE DIFFUSION MODEL

Researchers have explored a variety of representations to train text-to-image diffusion models, including pixel-diffusion models (Ramesh et al., 2022; Saharia et al., 2022; Balaji et al., 2022), latent diffusion models (Rombach et al., 2022; Dai et al., 2023), and token-based generative transformers (Chang et al., 2023; Sun et al., 2024; Li et al., 2024). Pixel-diffusion models directly generate images in the pixel space, but due to computational constraints, they typically first generate images at a lower resolution (e.g., $64 \times 64$) and then upsample them (sometimes multiple times) to achieve the target resolution in a cascade fashion (Saharia et al., 2022).

Latent Diffusion Models (LDMs), on the other hand, employ a pretrained autoencoder (Rombach et al., 2022) to compress the spatial dimensions of the image to be generated, typically by a factor of $8 \times 8$, while moderately increasing the channel dimension from 3 (RGB) to 4. This approach significantly enhances training efficiency compared to pixel diffusion models, thereby facilitating various applications such as high-resolution (Chen et al., 2023) and real-time image generation (Kohler et al., 2024; Wimbauer et al., 2024). Early LDM models use convolutional U-Nets as the backbone diffusion model, such as LDM1.5 (Rombach et al., 2022) and Emu (Dai et al., 2023). Recently, the field has been dominated by diffusion transformers (DiTs), such as SD3 (Esser et al., 2024) and PixArt-$\alpha$ (Chen et al., 2023). PixArt-$\alpha$ incorporates cross-attention modules into DiT and trained the model on high-aesthetic data in its final training stage. However, all of these LDM methods are still trained in latent space, which might suffer from loss of details and artifacts due to low spatial resolution.

In this paper, we propose a novel approach to refining image quality in diffusion models by deploying a pixel-space objective function in the post-training stage. Our method does not depend on a particular diffusion model type and works equally well for both U-Nets and DiTs.

### 2.2 SUPERVISED QUALITY FINE-TUNING (SFT)

Supervised fine-tuning is crucial to the success of modern LLMs (Zhou et al., 2024; Touvron et al., 2023; Achiam et al., 2023). In image and vision, Dai et al. (2023) proposes using a small set of extremely high-quality images to fine-tune a pretrained LDM model, resulting in significant improvements to the visual quality of generated images without sacrificing text-image alignment. Betker et al. (2023) and Segalis et al. (2023) propose using captions rewritten by vision language models to facilitate better learning, including during SFT. However, none of these proposed methods explored the representation space in which the model was fine-tuned. In this paper, we propose supplementing the regular supervised fine-tuning loss with a pixel-space objective function. We experimented with two different models: a replacement-trained U-Net LDM-1.5 (Rombach et al., 2022) and a DiT model. Our results show that when fine-tuning on a small high-quality dataset, our proposed method can significantly improve generation quality and visual flaws.

### 2.3 HUMAN PREFERENCE BASED POST TRAINING

Reinforcement learning represents another popular type of post-training technique. The seminal work of Schulman et al. (2017) makes the policy gradient method practical. Rafailov et al. (2024) proposes doing direct preference optimization (DPO) with a reference model to improve the model quality on language models. Wallace et al. (2024) and Black et al. (2023) extend DPO to diffusion models. DPO optimizes diffusion models on paired human preference data by implicitly estimating a reward model. Liang et al. (2024) proposes doing step-aware DPO. Meng et al. (2024), on the other hand, remove the reference model to make reinforcement learning more direct and effective. In this paper, we show that our proposed pixel-space objective also works well for reward-based post-training.

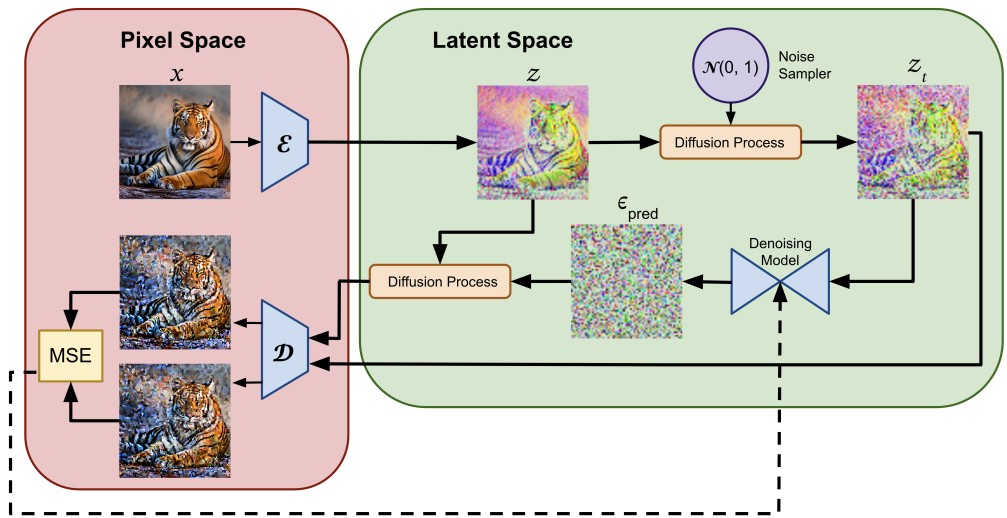

Figure 2: **Supervised fine-tuning with pixel-space loss.** During fine-tuning, we decode the latent representation back to the pixel space and add a supervision in the output image resolution.

## 3 METHOD

Given an image $x \in \mathbb{R}^{H \times W \times 3}$ in RGB space, LDMs use an autoencoder $\mathcal{E}$ that encodes $x$ into a latent representation $z = \mathcal{E}(x)$. The decoder $\mathcal{D}$ then reconstructs the image from the latent, giving $\tilde{x} = \mathcal{D}(z) = \mathcal{D}(\mathcal{E}(x))$, where $z \in \mathbb{R}^{h \times w \times c}$.

### 3.1 SUPERVISED PIXEL-SPACE FINE-TUNING

By denoising a normally distributed variable step-by-step, LDMs learn a data distribution $p_\theta(x)$. Therefore, they can be understood as a series of denoising autoencoders $\epsilon_\theta(z_t, t)$; $t = 1 \dots T$ that are trained to predict the denoised variant of their input $z_t$ where $z_t$ is the noisy version of latent input $z$ at time $t$, $\epsilon$ is the original noise added to get $z_t$, and $\epsilon_\theta(z_t, t)$ is the predicted noise. Furthermore, the noise added to $z_{t-1}$ to get $z_t$ is Gaussian with variance $\beta_t$. The standard objective function for LDMs is:

$$L_{SFT}^{latent} := \mathbb{E}_{\mathcal{E}(x), \epsilon \sim \mathcal{N}(0,1), t} \left[ \|\epsilon - \epsilon_\theta(z_t, t)\|_2^2 \right]. \tag{1}$$

Instead of working only in the latent space $\mathbb{R}^{h \times w \times c}$, we propose a loss function that incorporates the pixel space $\mathbb{R}^{H \times W \times 3}$ in the objective function. This is achieved by adding the noise $\epsilon$ to the latent image $z$ through the forward diffusion process $x_t = \sqrt{\bar{\alpha}_t} x_0 + \sqrt{1 - \bar{\alpha}_t}\epsilon$, where $\alpha_t = 1 - \beta_t$ and $\bar{\alpha}_t = \prod_{i=1}^t \alpha_i$, and decoding it back to the pixel space. The objective function then becomes

$$L_{SFT}^{pixel} := \mathbb{E}_{\mathcal{E}(x), \epsilon \sim \mathcal{N}(0,1), t} \left[ \|\mathcal{D}(\sqrt{\bar{\alpha}_t} z + \sqrt{1 - \bar{\alpha}_t}\epsilon) - \mathcal{D}(\sqrt{\bar{\alpha}_t} z + \sqrt{1 - \bar{\alpha}_t}\epsilon_\theta(z_t, t))\|_2^2 \right]. \tag{2}$$

We combine the latent objective, Equation 1, with the pixel objective, Equation 2, to obtain an objective that uses both the latent and pixel space, weighted by hyper-parameter $\lambda$.

$$L_{SFT} := L_{SFT}^{latent} + \lambda L_{SFT}^{pixel}. \tag{3}$$

### 3.2 PIXEL-SPACE FINE-TUNING USING REWARD MODELING

Define $x^w$ and $x^l$ to be the "winning" and "losing" samples from human annotations, then $z^w = \mathcal{E}(x^w)$ and $z^l = \mathcal{E}(x^l)$ represent the "winning" and "losing" samples in the latent space. Unlike regular supervised fine-tuning, fine-tuning with DPO utilizes a reference distribution $p_{\text{ref}}(x)$ and hyperparameter $\beta$ for regularization. Fine-tuning now aims to learn $p_\theta$, which is aligned to human preferences, while still remembering $p_{\text{ref}}$. The reward modeling objective for fine-tuning takes the form:

$$L_{dpo}^{latent} := - \mathbb{E}_{\mathcal{E}(x), \epsilon \sim \mathcal{N}(0,1), t} \log \sigma(-\beta(\|\epsilon^w - \epsilon_\theta(z_t^w, t)\|_2^2 -$$
$$\|\epsilon^w - \epsilon_{\text{ref}}(z_t^w, t)\|_2^2 - (\|\epsilon^l - \epsilon_\theta(z_t^l, t)\|_2^2 - \|\epsilon^l - \epsilon_{\text{ref}}(z_t^l, t)\|_2^2))). \tag{4}$$

Inspired by Meng et al. (2024), we remove the reference model and simplify the objective to

$$L_{simpo}^{latent} := -\mathbb{E}_{\mathcal{E}(x),\epsilon\sim\mathcal{N}(0,1),t} \log \sigma \left(-\beta(\|\epsilon^w - \epsilon_\theta(z_t^w, t)\|_2^2 - (\|\epsilon^l - \epsilon_\theta(z_t^l, t)\|_2^2))\right). \tag{5}$$

Similar to supervised fine-tuning, we also incorporate calculations in the pixel space into our reward modeling. and define the pixel-space objective as

$$\begin{aligned} L_{simpo}^{pixel} := -\,&\mathbb{E}_{\mathcal{E}(x),\epsilon\sim\mathcal{N}(0,1),t} \log \sigma(\\ &-\beta((\|\mathcal{D}(\sqrt{\bar{\alpha}_t}z^w + \sqrt{1-\bar{\alpha}_t}\epsilon^w) - \mathcal{D}(\sqrt{\bar{\alpha}_t}z^w + \sqrt{1-\bar{\alpha}_t}\epsilon_\theta(z_t^w, t))\|_2^2\\ &-(\|\mathcal{D}(\sqrt{\bar{\alpha}_t}z^l + \sqrt{1-\bar{\alpha}_t}\epsilon^l) - \mathcal{D}(\sqrt{\bar{\alpha}_t}z^l + \sqrt{1-\bar{\alpha}_t}\epsilon_\theta(z_t^l, t))\|_2^2))). \end{aligned} \tag{6}$$

Combining latent and pixel terms and weighted by a constant $\mu$, we get

$$L_{reward} = L_{simpo}^{latent} + \mu L_{simpo}^{pixel}. \tag{7}$$

## 4 EXPERIMENT

We conduct a comprehensive qualitative and quantitative analysis, as well as ablation studies to show that our proposed loss function outperforms the regular latent space loss in both supervised fine-tuning and preference-based post-training.

### 4.1 HUMAN EVALUATION

Like many recent studies, we found that a rigorous and independent human evaluation process is the most reliable way to evaluate different models. Commonly used metrics such as the FID score do not correlate well with human preference (Dai et al., 2023; Podell et al., 2023; Kirstain et al., 2023).

We contracted a team of paid and independent annotators who do not have contexts of our project to evaluate the generated images. We conducted A/B comparisons on visual flaws and visual appeal, as well as standalone evaluations on text alignment. We use a 600-prompt list in the GenAI MAGIC challenge for the evaluation (Tsai et al., 2024), where each example is annotated by at least 5 people and the majority decision is used.

**Visual flaws.** The annotators were presented with two images side-by-side, generated by two different models, without the prompt. The annotators were trained to identify major flaws (e.g., displaced body parts) and minor flaws (distorted eyes), and are asked to choose from "left wins", "tie" and "right wins".

**Visual appeal.** Similar to the visual flaws task, but the annotators are asked to compare which image is more aesthetically pleasing. Annotators were instructed to reject any examples where one image was photo-realistic and the other was stylized (e.g., a cartoon).

**Text alignment.** We predefined a list of binary questions for each prompt and asked annotators to answer yes or no. We calculated the text alignment rate by aggregating the results across all questions. For example, for the prompt "a cat and a dog", the annotators are asked "is there one dog", "is there one cat", and "are there other animals present".

### 4.2 EXPERIMENTAL SETUP

**Baseline.** We tested our model on three models: 1) A 0.6B parameter DiT model with standard transformer and cross attention blocks and trained with high quality data in an "annealing" stage after pretraining to generate high quality images, 2) A 0.86B parameter U-Net with LDM1.5 architecture (Rombach et al., 2022), replacement-trained on 300M Shutterstock data without quality tuning, which thus generates lower-quality images without prompt engineering and 3) A larger U-Net based Emu model (Dai et al., 2023) that has been quality fine-tuned, and generates the highest quality images among the three.

**Supervised fine-tuning.** We curated a small, high-quality dataset of 1816 images for fine-tuning, following the practice of Dai et al. (2023). Since Emu is already quality fine-tuned, we focused on replacement-trained LDM1.5 and DiT. For supervised fine-tuning with a small dataset, style

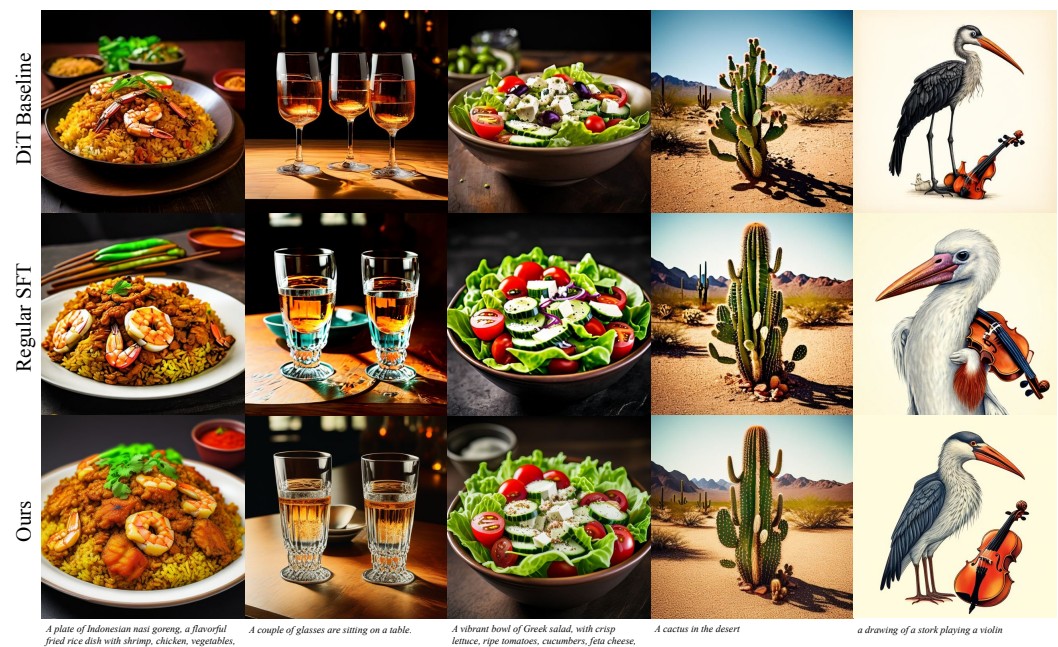

DiT Baseline

Regular SFT

Ours

*A plate of Indonesian nasi goreng, a flavorful fried rice dish with shrimp, chicken, vegetables, and a hint of sambal sauce.*

*A couple of glasses are sitting on a table.*

*A vibrant bowl of Greek salad, with crisp lettuce, ripe tomatoes, cucumbers, feta cheese, and a drizzle of olive oil.*

*A cactus in the desert*

*a drawing of a stork playing a violin*

Figure 3: **SFT with pixel-space loss: DiT.** Fine-tuning with our method improves visual quality and reduces flaws. Zoom in to see details.

consistency is crucial. We found that using a hand-picked set of generated images from a high-quality model is sufficient. Examples of our curated fine-tune data are in Appendix Figure 6.

**Preference-based fine-tuning.** We conducted experiments on the higher-quality U-Net Emu and DiT models. For each model, we generate 5 images per prompt and ask annotators to select a positive and negative pair. In instances where visual flaws and visual quality conflicted, we prioritize the image with fewest visual flaws as the positive example.

**Implementation details.** We run all experiments at $512 \times 512$ resolution for LDM1.5 and DiT, and $768 \times 768$ for Emu, using Adam optimizer with weight decay of $5e-6$. For preference-based fine-tuning, we set $\lambda = \mu = 8.0$ for DiT and 2.0 for Emu to balance the pixel loss magnitude and latent loss magnitude, running 100 epochs. For supervised fine-tuning, we empirically use 140 epochs. We ablate the hyper-parameters such as learning rate and batch size for each model and choose the best ones. Each experiment took 1-8 hours on 8 H100 GPUs to fine-tune one model. During inference, we use the standard DDIM solver with 50 steps with classifier-free guidance.

### 4.3 EXPERIMENTAL RESULTS

**Supervised Fine-tuning.** After supervised fine-tuning, our proposed loss improved visual flaws win rate from $17.7\%$ to $64.2\%$, and visual quality win rate from $47.9\%$ to $64.7\%$, compared to regular fine-tuning against the DiT baseline. When directly comparing the model fine-tuned with our loss to the model fine-tuned with the regular latent space loss, ours showed a $32.8\%$ vs $9.3\%$ win rate on visual flaws and $34.8\%$ vs $16.6\%$ win rate on visual appeal. We also found that supervised fine-tuning did not affect text alignment, with correct alignment rates of $74.0\%$, $74.3\%$ and $74.0\%$ for baseline, regular latent fine-tuning, and our fine-tuning respectively, (Table 1), all within the margin of error for the annotations. Qualitative examples in Figure 3 demonstrate that our method generates fewer artifacts and much better fine details.

Table 2 show the results for LDM1.5 (replacement trained). Our proposed SFT with pixel loss still improves over regular latent SFT in head-to-head comparisons (last row). Comparing with the DiT experiments, the smaller difference between pixel and regular SFT when compared to the baseline is due to LDM1.5's lower image generation quality, as it was only replacement trained with Shutterstock data. Therefore, both regularly SFT-ed and pixel SFT-ed models significantly

Table 1: Pixel space loss improves supervised fine-tuning: DiT.

| Model A | Model B | Visual Flaws | | | Visual Appeal | | | Text alignment | |
| | | A Wins | Tie | B Wins | A Wins | Tie | B Wins | Model A | Model B |
| --- | --- | --- | --- | --- | --- | --- | --- | --- | --- |
| Regular SFT | Baseline | 17.7% | 74.0% | 8.3% | 47.9% | 19.3% | 32.8% | 74.3% | 74.0% |
| Ours | Baseline | 64.2% | 26.5% | 9.3% | 64.7% | 20.6% | 14.8% | 74.0% | 74.0% |
| Ours | Regular SFT | **32.8**% | 57.8% | 9.3% | **34.8**% | 48.6% | 16.6% | 74.0% | 74.3% |

Table 2: Pixel space loss improves fine-tuning: Replacement trained LDM1.5.

| Model A | Model B | Visual Flaws | | | Visual Appeal | | | Text alignment | |
| | | A Wins | Tie | B Wins | A Wins | Tie | B Wins | Model A | Model B |
| --- | --- | --- | --- | --- | --- | --- | --- | --- | --- |
| Regular SFT | Baseline | 75.0% | 13.7% | 11.3% | 79.6% | 6.9% | 13.5% | 72.3% | 61.4% |
| Our SFT | Baseline | 74.2% | 16.7% | 9.1% | 75.2% | 10.8% | 14.0% | 72.1% | 61.4% |
| Our SFT | Regular SFT | **29.3**% | 46.7% | 24.0% | **41.5**% | 24.0% | 34.4% | 72.1% | 72.3% |

Table 3: Our proposed pixel-space objective also significantly improves DPO on the DiT model.

| Model A | Model B | Visual Flaws | | | Visual Appeal | | | Text alignment | |
| | | A Wins | Tie | B Wins | A Wins | Tie | B Wins | Model A | Model B |
| --- | --- | --- | --- | --- | --- | --- | --- | --- | --- |
| Regular DPO | Baseline | 27.5% | 63.7% | 8.8% | 48.3% | 39.7% | 12.0% | 72.8% | 74.0% |
| Ours | Baseline | 43.3% | 43.7% | 13.0% | 61.2% | 24.8% | 14.0% | 75.7% | 74.0% |
| Ours | Regular DPO | **31.0**% | 49.3% | 19.7% | **43.7**% | 30.4% | 25.9% | 75.7% | 72.8% |

outperform the baseline in terms of visual quality and flawlessness, and supervised fine-tuning in this case, focuses on learning the overall style and aesthetics of the fine-tune images instead of fine details. See Figures 1 for qualitative examples. Notably, text alignment also improved with SFT, consistent with findings from Dai et al. (2023).

**Preference-based fine-tuning.** Our method also demonstrates exceptional performance in reward-based fine-tuning, generating significantly more impressive details than the baselines. The results are best demonstrated qualitatively in Figure 4 and 5. Quantitatively, as shown in Table 3, compared to regular DPO, our proposed pixel objective function improves the win rate from 27.5% to 43.3% for visual flaws and 48.3% to 61.2% for visual appeal when evaluated against the baseline DiT. When doing head-to-head comparison between our method and regular DPO, we achieve win rates of 31.0% vs 19.7% on visual flaws and 43.7% vs 25.9% on visual appeal. Although unintended, our method also improves text alignment by 2.9%.

With U-Net based Emu (Figure 5), the baseline model already generates higher-quality flawless images in most cases. Therefore, the flaw comparison will result in ties in majority of the cases. Despite this, our proposed method still manages to improve visual flaws win rate from 2.7% to 16.0% and visual appeal from 36.3% to 42.2% as shown in Table 4.

Table 4: Pixel-space objective also improves DPO on Emu.

| Model A | Model B | Visual Flaws | | | Visual Appeal | | | Text alignment | |
| | | A Wins | Tie | B Wins | A Wins | Tie | B Wins | Model A | Model B |
| --- | --- | --- | --- | --- | --- | --- | --- | --- | --- |
| Regular DPO | Baseline | 2.7% | 95.3% | 2.0% | 36.3% | 34.9% | 28.8% | 89.5% | 89.2% |
| Ours | Baseline | 16.0% | 75.5% | 8.5% | 42.2% | 31.4% | 26.5% | 89.3% | 89.2% |
| Ours | Regular DPO | **18.3**% | 70.2% | 11.5% | **13.7**% | 80.0% | 6.3% | 89.3% | 89.5% |

**Additional qualitative examples.** We provide additional qualitative examples for each experiment above in the Appendix.

### 4.4 ABLATION STUDIES

**Latent vs Pixel vs Pixel+Latent Loss.** When only using the pixel space loss during supervised fine-tuning, we noticed that the resulting images had very clear details in the main focus of the image, but the background tends to be overly blurred as if they are photographs taken with an extremely narrow depth of field. As shown in Table 5, using pixel space alone also significantly improves

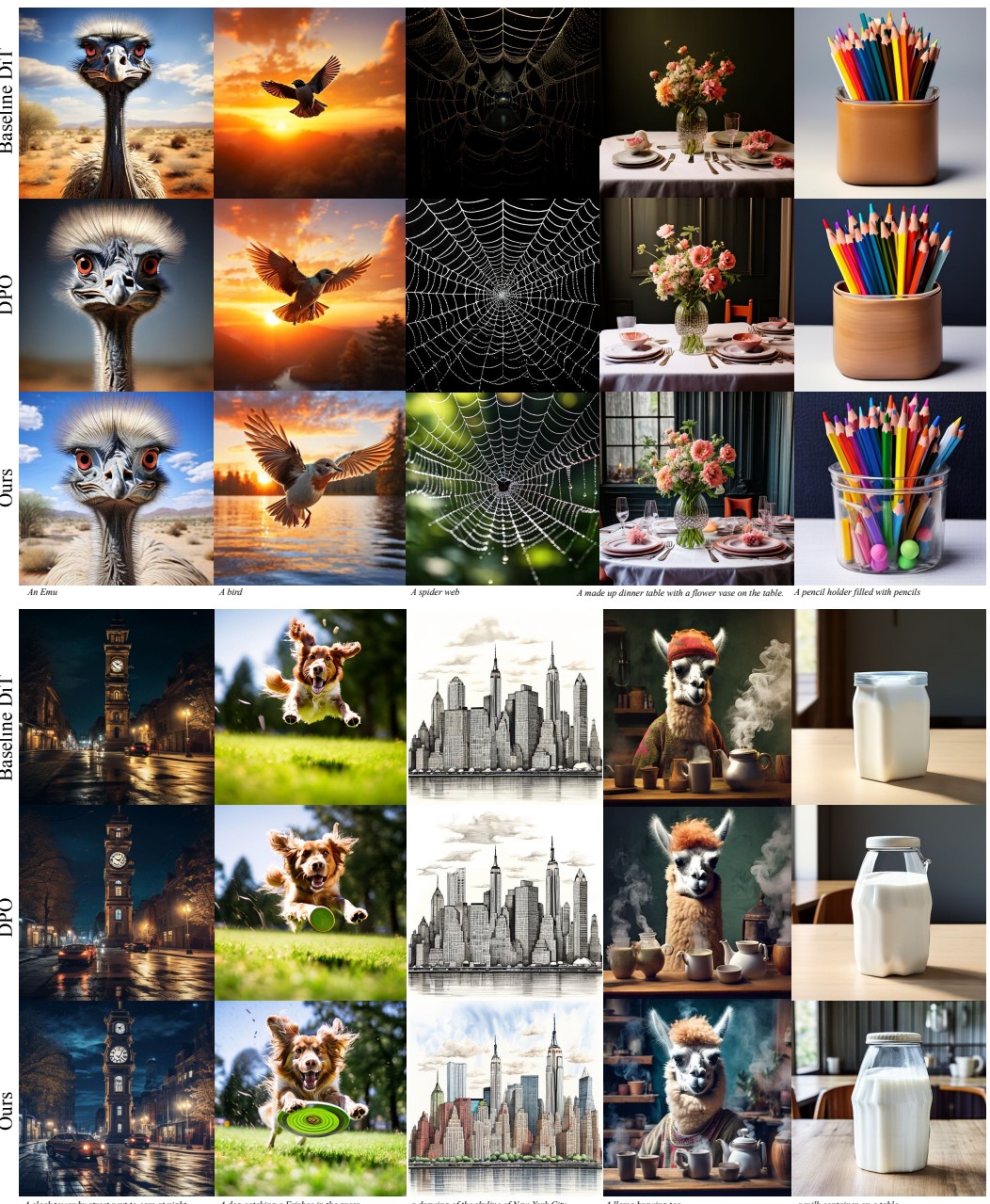

Figure 4: **Preference-based post-training with pixel-space loss: DiT.** The model trained with our proposed objective function generates more stunning fine details and fewer artifacts.

visual flaws, helping the images look more realistic and crisp. However by combining it with latent loss, we are able to significantly improve the visual appeal as well, resulting in images with more stunning details, especially in the background.

**Decoding Methodology.** Intuitively, to obtain as much image quality and details as possible, one may consider regressing to an objective function that compares the output to the original starting image in the pixel space. This involves two steps: first, transforming the predicted noise in the latent space back to $t = 0$, then decoding into the pixel space using equation

$$x_0 = \mathcal{D}(z_0) = \mathcal{D}\left(\frac{1}{\sqrt{\bar{\alpha}_t}}(z_t - \sqrt{1 - \bar{\alpha}_t}\epsilon_\theta(z_t, t))\right). \tag{8}$$

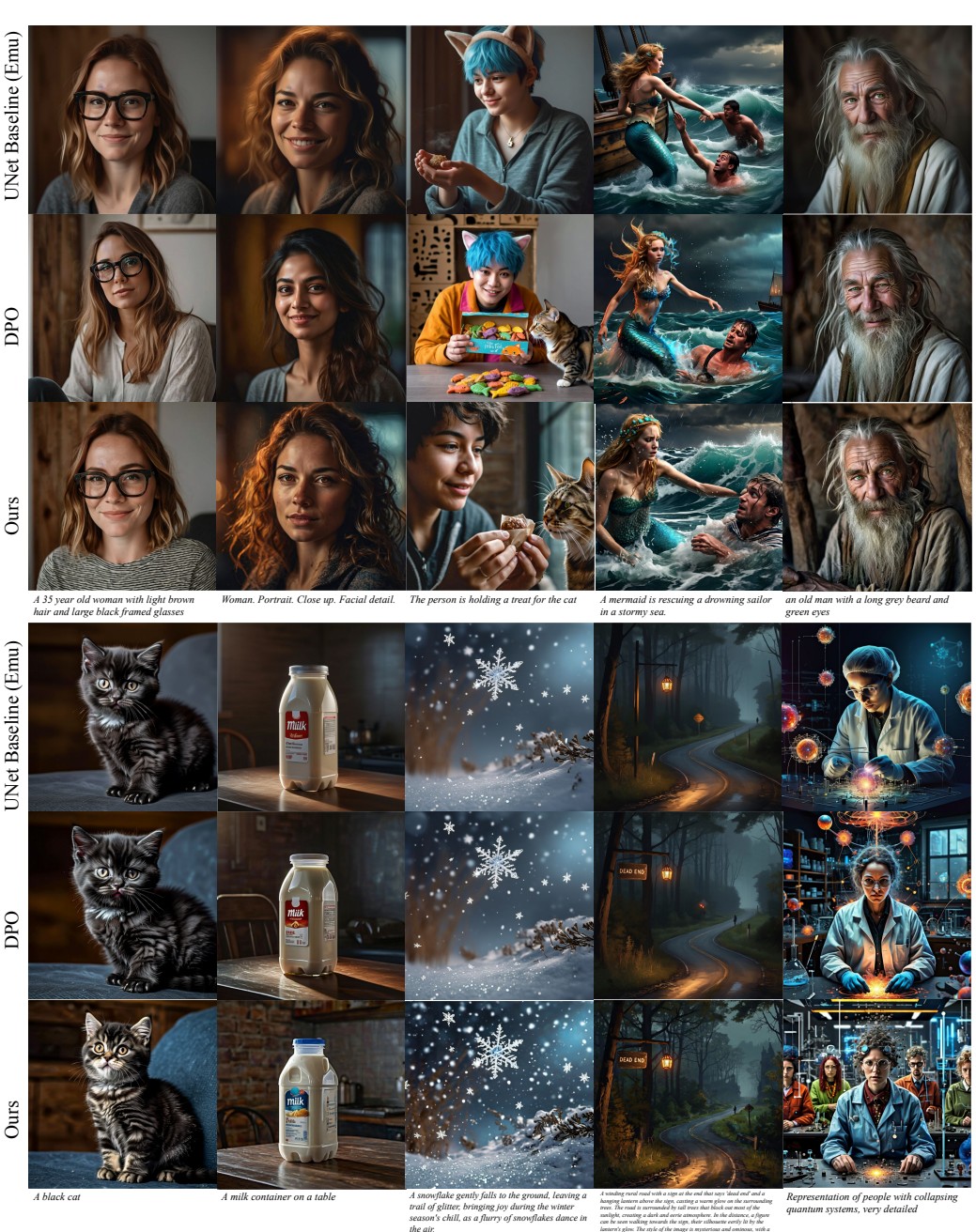

Figure 5: **Preference-based post-training with pixel-space loss: Emu.** Similar to DiT, the Emu model fine-tuned using our loss generates even better details, despite how the baseline Emu already generates good quality images with rich details. Zoom in to see the improvements.

Although this method seems like it would be optimal in generating the detail and high quality desired in the final image, the transformed $x_0$ has a greater variance for larger timesteps, causing the generated images to be blurry and fuzzy (Appendix Figure 7) since the fine-tuning process was trying to correct for the estimation error of $z_0$.

Based on this finding, we propose comparing the output directly with the sample in the pixel space at the current timestep to eliminate the unwanted variations in the previous method, using Equation 2. Empirically, we have found that even though noisy images are out-of-distribution for the autoencoder, it still does surprisingly well in reconstructing them.

Table 5: Ablation study: Supervised fine-tuning method: Latent, Pixel, vs Latent + Pixel (Ours).

| Model A | Model B | Visual Flaws | | | Visual Appeal | | |
|---|---|---|---|---|---|---|---|
| | | A Wins | Tie | B Wins | A Wins | Tie | B Wins |
| Latent only | Baseline DiT | 17.7% | 74.0% | 8.3% | 46.1% | 20.7% | 33.2% |
| Pixel only | Baseline DiT | 43.8% | 42.7% | 13.5% | 44.8% | 21.7% | 33.5% |
| Ours (Latent and Pixel) | Baseline DiT | **64.2%** | 26.5% | 9.3% | **63.8%** | 21.5% | 14.7% |
| Ours (Latent and Pixel) | Latent only | **32.8%** | 57.8% | 9.3% | **34.8%** | 48.6% | 16.6% |
| Ours (Latent and Pixel) | Pixel only | **22.7%** | 60.5% | 16.8% | **47.5%** | 38.5% | 14.0% |

**Reference Model in Reward Modeling.** Traditionally, DPO (Rafailov et al., 2024; Wallace et al., 2024) utilizes a reference model, but recently Meng et al. (2024) proposed removing the reference model (SimPO) in LLMs and showed strong performance. Here, we tested out different combinations of using latent-space loss and pixel-loss with and without the reference model, as shown in Table 6. We show that simply adding our proposed pixel loss term can already significantly improve the visual flaws metric using the standard DPO method ($53.0\%$ vs $27.5\%$ win rate compared to baseline DiT model). However, by incorporating SimPO, we achieve both significant improvement on visual flaws and visual appeal (improving from $27.5\%$ to $43.3\%$ in visual flaws and $47.2\%$ to $61.2\%$ in visual appeal). To the best of our knowledge, this is also the first paper that demonstrates that the recent success of SimPO can also be extended to diffusion models.

Table 6: Ablation Study: Reward modeling variations vs Baseline DiT

| Model | Visual Flaws | | | Visual Appeal | | |
|---|---|---|---|---|---|---|
| | Win | Tie | Lose | Win | Tie | Lose |
| DPO latent (baseline) | 27.5% | 63.7% | 8.8% | 47.2% | 40.7% | 12.2% |
| DPO latent + DPO pixel | **53.0%** | 31.5% | 15.5% | 49.0% | 21.0% | 30.0% |
| DPO latent + SimPO pixel | 50.7% | 35.7% | 13.7% | 41.3% | 22.8% | 35.8% |
| SimPO latent + SimPO pixel (Proposed) | 43.3% | 43.7% | 13.0% | **61.2%** | 24.8% | 14.0% |

## 5 LIMITATIONS

**Limitations of Baseline Models.** Fine-tuning improvements are dependent on the quality of the original baseline model, and thus fine-tuning using pixel space loss may not always produce significant improvements. For example, if the original baseline model already has minimal flaws and high visual appeal, fine-tuning may not achieve many improvements. In contrast, if the original baseline foundation model generates significant structural flaws that require the global understanding of the image composition, fine-tuning with our loss alone may not help eliminate them.

**Limitations of Fine-tuning Dataset.** Fine-tuning is also dependent on the quality and composition of the dataset used. Our dataset was hand-curated and consisted of images that followed our definition of high quality and style. Changing the composition of this dataset would lead to different results.

**Limitations of Human Evaluation.** The images generated by the different models were evaluated by independent annotators. Although the annotators were trained on standards for visual flaws, visual appeal, and text alignment, these results may not fully reflect the real-world use of the models. Human evaluation is also inherently subjective and noisy in terms of aesthetics.

## 6 CONCLUSIONS

In this paper, we proposed a novel post-training objective function for latent diffusion models by incorporating a pixel-space loss with the commonly used latent-space fine-tuning loss. The resulting model shows noticeable improvement in visual flaws and visual appeal metrics in both supervised fine-tuning and preference-based post-training through rigorous human evaluations. The proposed objective function is simple and can be easily plugged into existing models such as DiT and U-Net.

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

# A APPENDIX

## A.1 FINE-TUNE DATASET

Here we show some examples of our supervised fine-tune dataset, which are selected images generated by Emu (Dai et al., 2023).

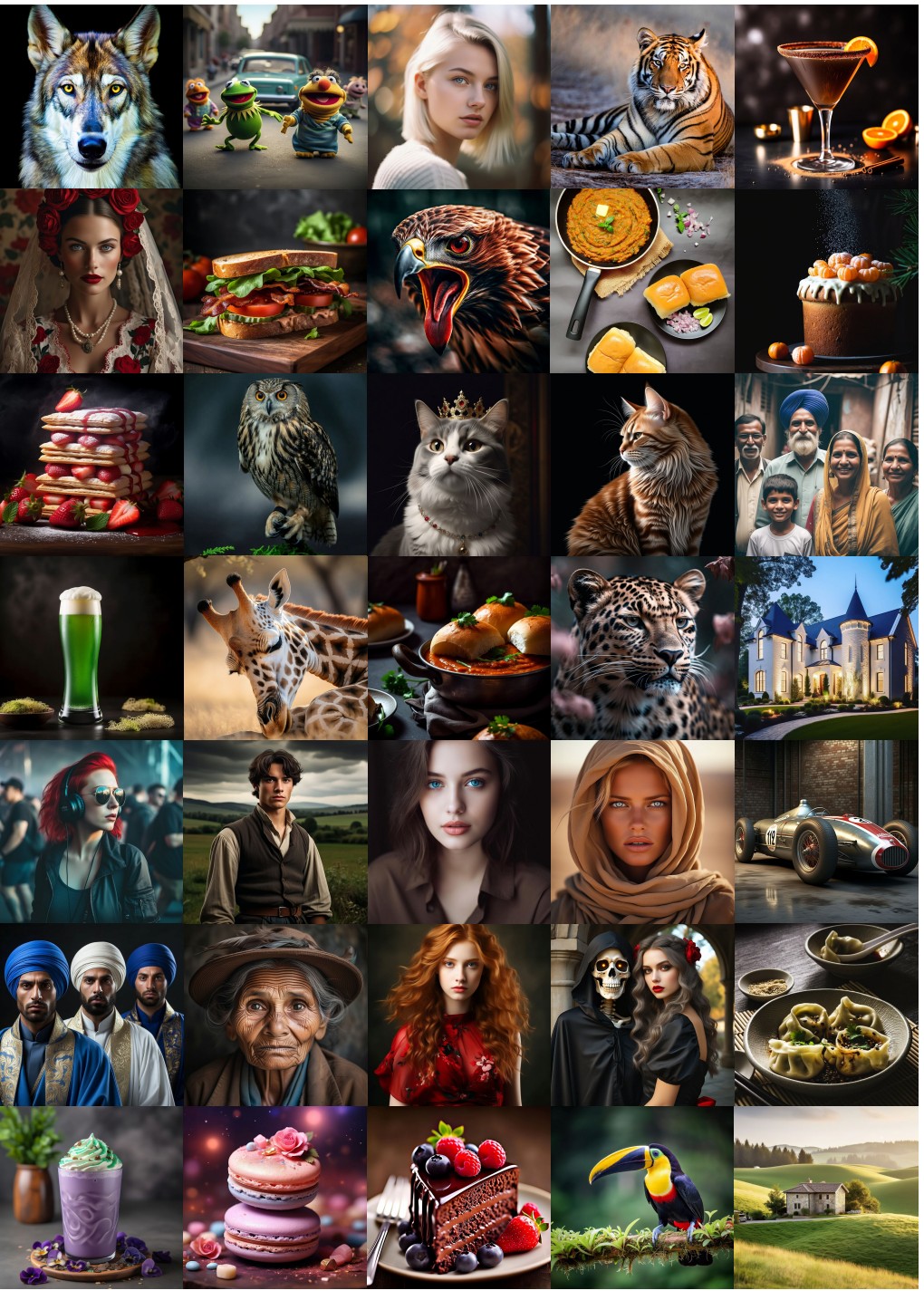

Figure 6: **Fine-tune dataset.** Selected images in our supervised fine-tune dataset.

## A.2 DECODING METHODOLOGY

An alternative decoding methodology would be to transform the latent space back to $t = 0$, and then decode it into the pixel space to obtain $x_0$, as discussed in Section 4.4. Figure 7 shows that if one follows Equation 8 to decode back to $t = 0$, it leads to blurrier images for larger timesteps. Therefore, we chose to decode back directly at the present timestep, as discussed in Section 4.4 using Equation 2.

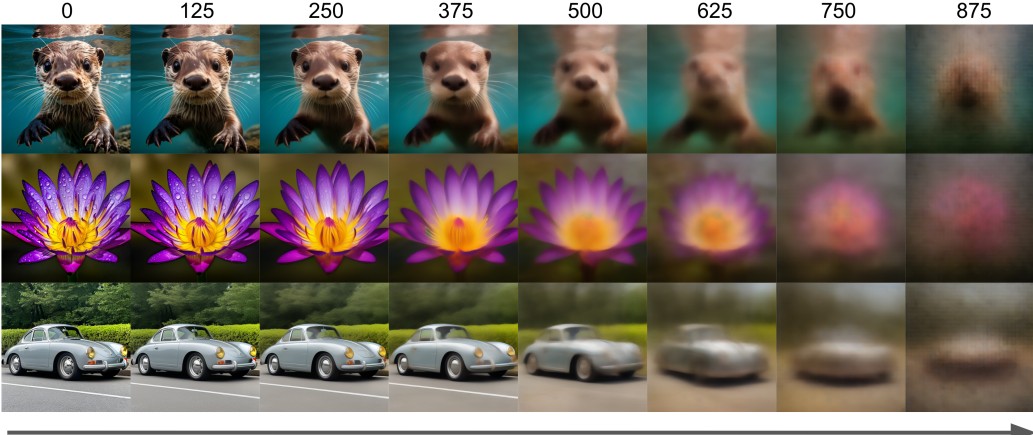

Figure 7: **Decoding methodology.** Transforming images back to $t_0$ causes the decoded images to be blurrier for larger timesteps. Therefore, we choose to decode at the present timestep.

A.3 MORE EXAMPLES

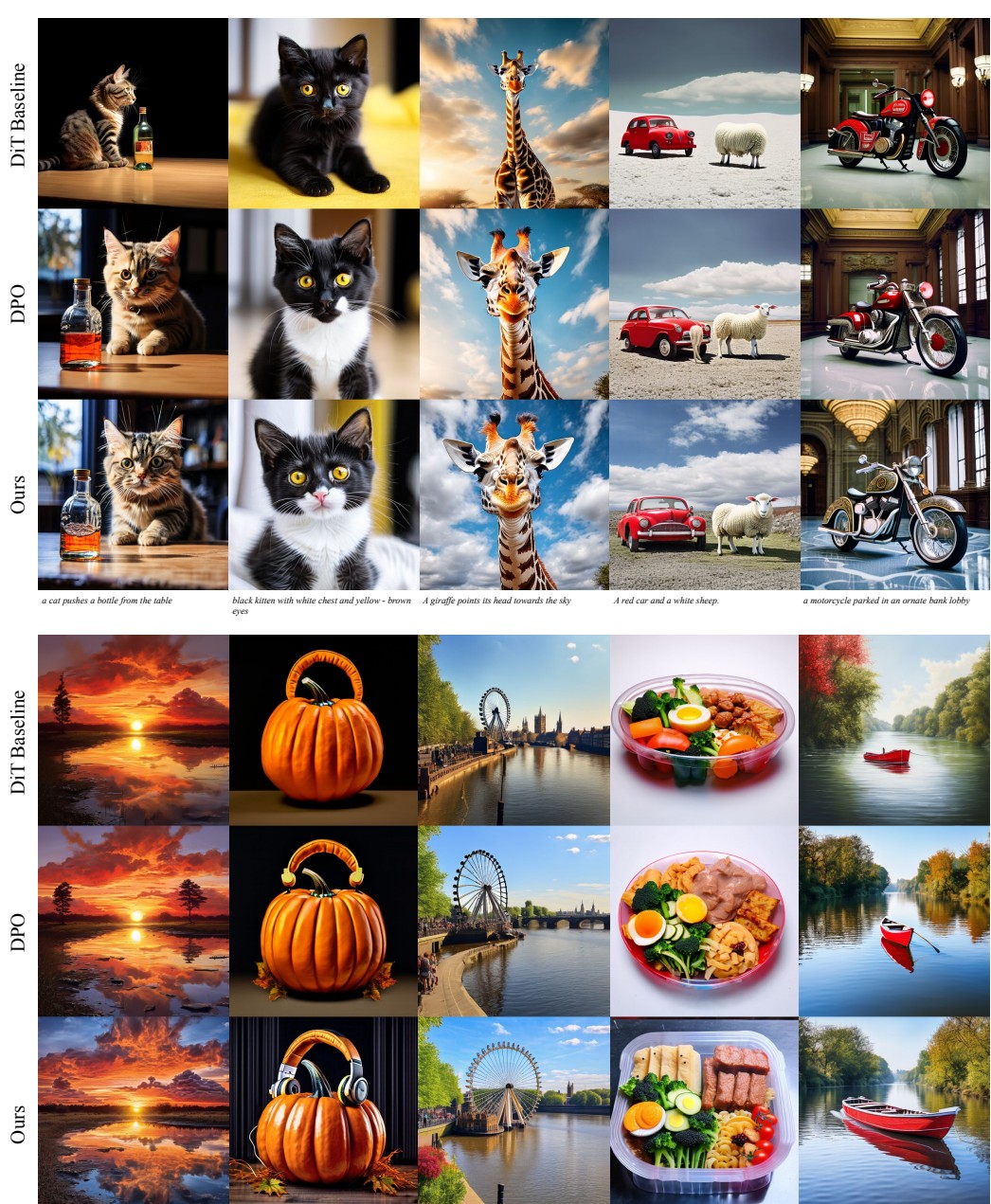

Figure 8: **More examples: DiT with preference-based fine-tuning.** We provide more examples here to demonstrate the improvements in high-frequency details and visual flaws.

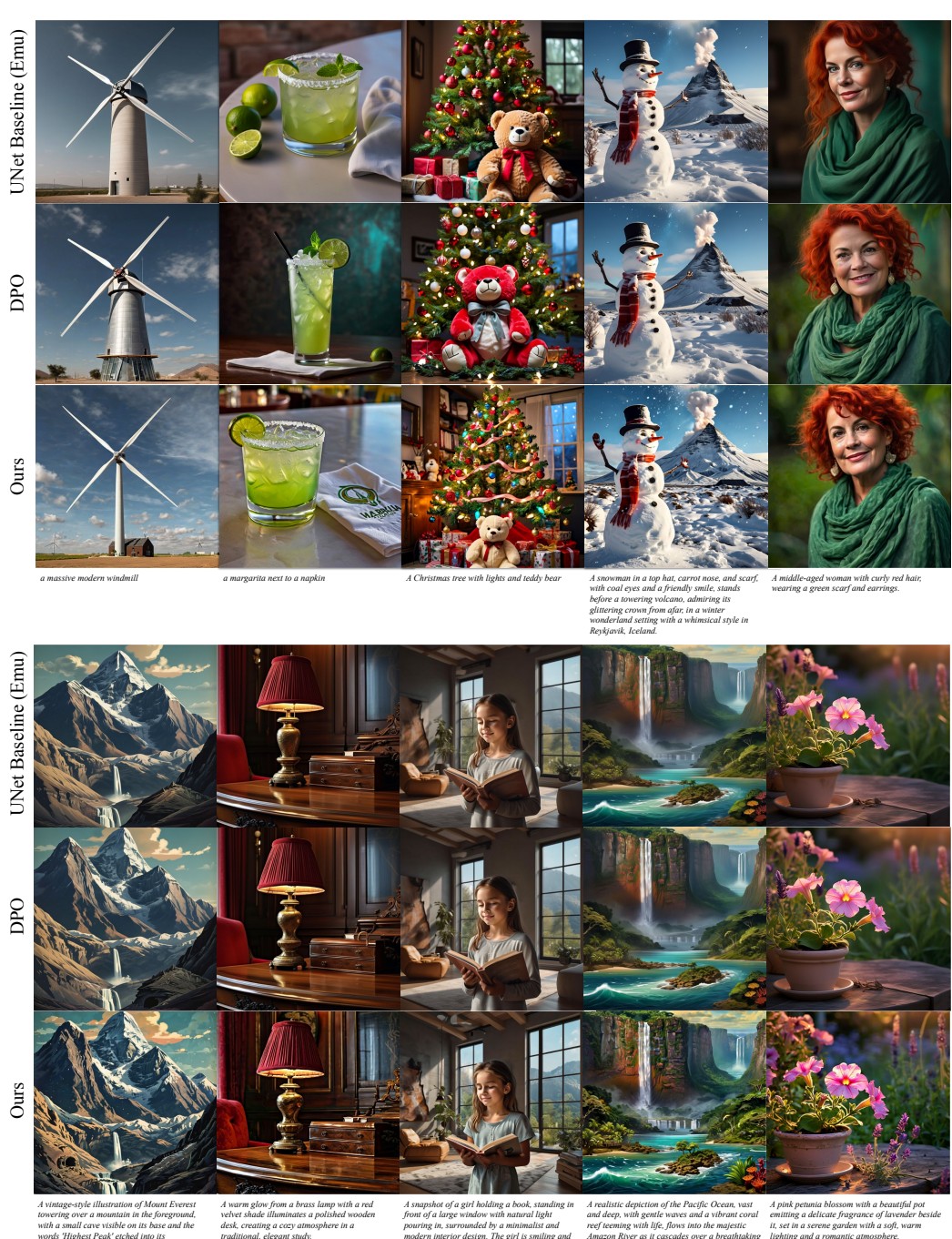

Figure 9: **More examples: U-Net Emu with preference-based fine-tuning.** Similar to the DiT model, we also observe improvements for Emu, despite how the baseline model already generates quite high quality images in most cases.

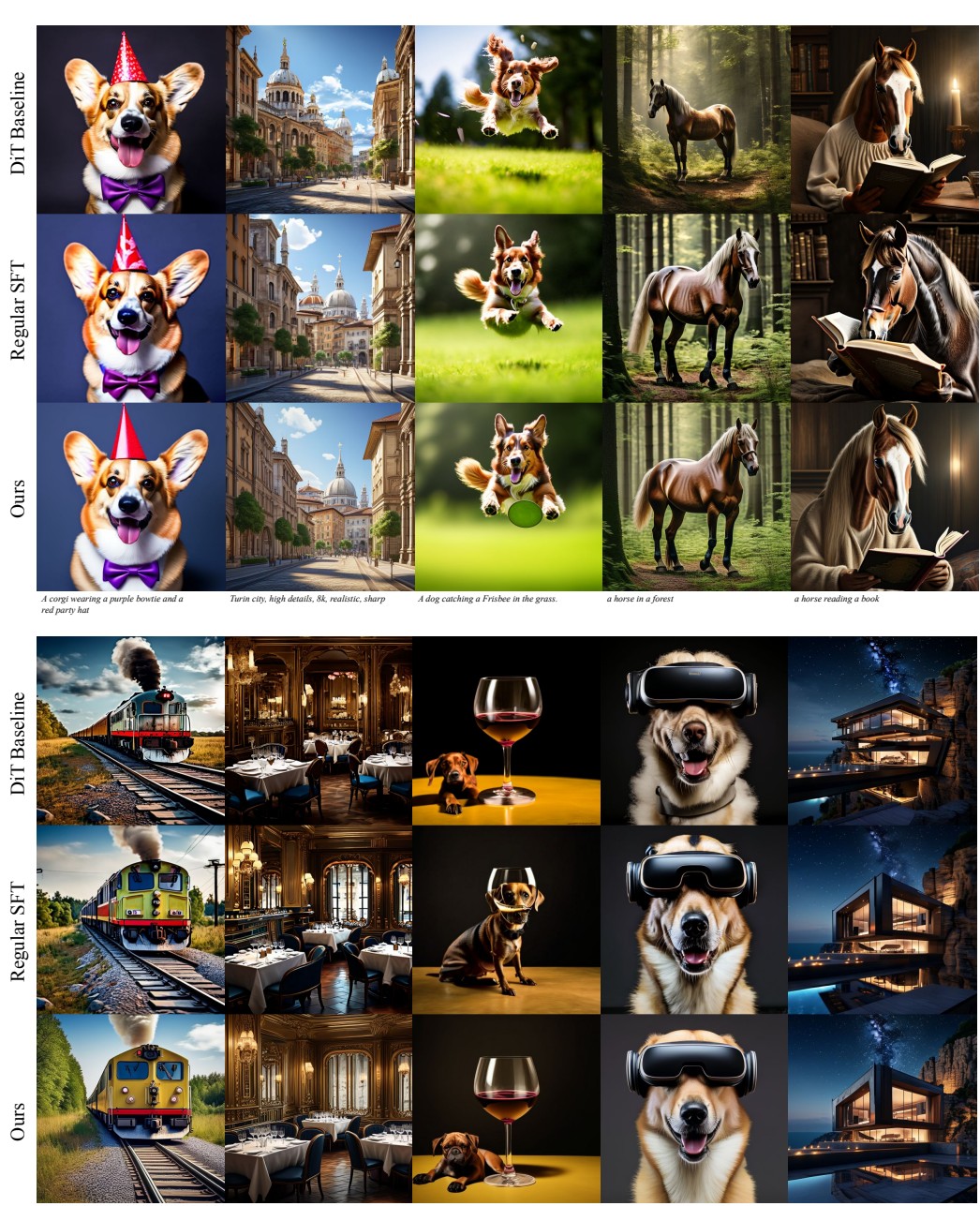

Figure 10: **More examples: DiT with supervised fine-tuning.** The model fine-tuned with our objective function often generates sharper images, which is best appreciated when zooming in. The model fine-tuned with regular latent loss, on the other hand, is typically blurrier, such as the animals' furs. Also as shown in many examples here, our model tends to generate fewer flaws.

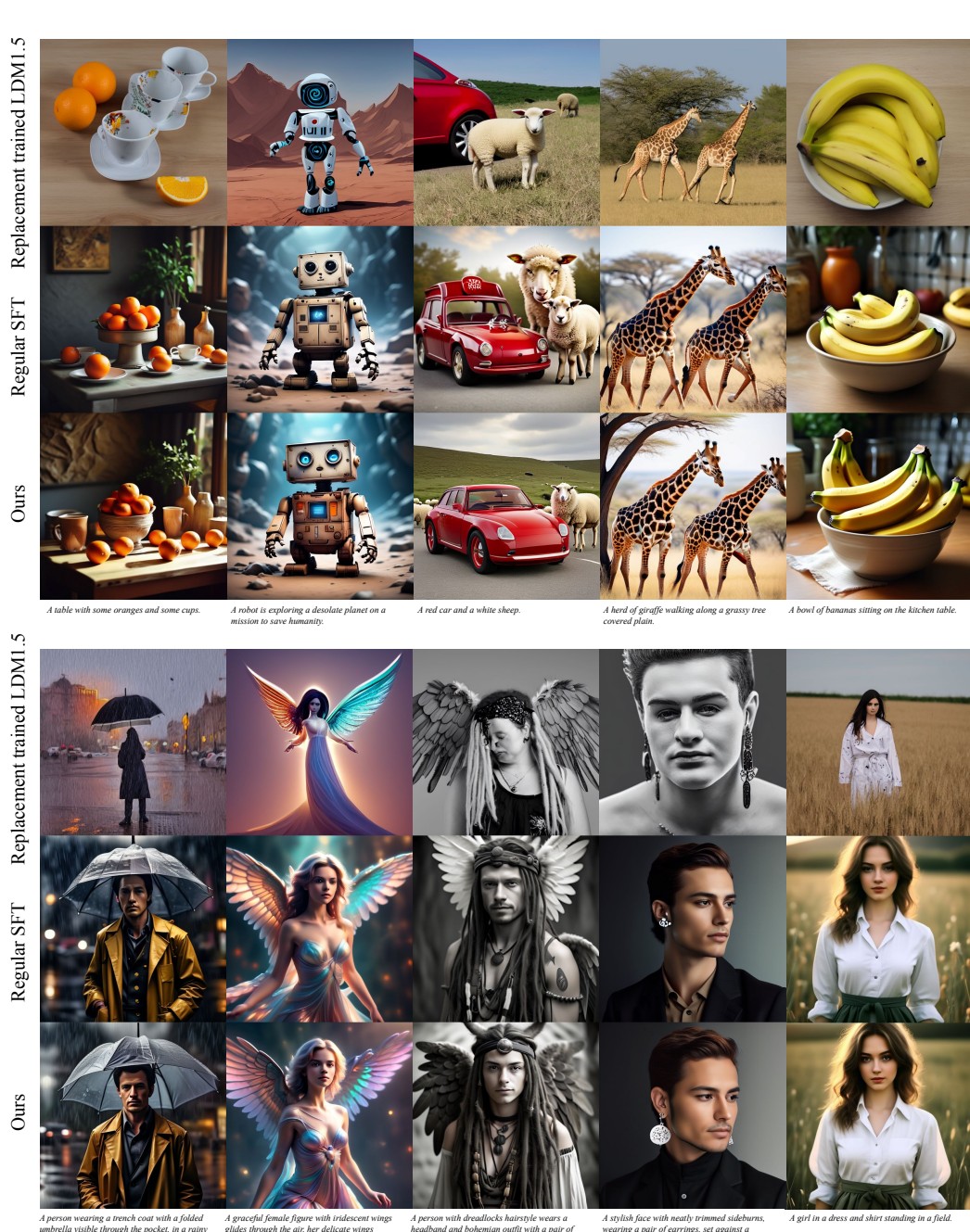

Figure 11: **More examples: U-Net LDM1.5 with supervised fine-tuning.** The baseline replacement-trained LDM1.5 often generates images with bad composition and noticeable artifacts. Supervised fine-tuning alone helps improve the image quality significantly. Our loss further improves visual quality and flaws. Again, readers can better appreciate the improvements when zooming in to notice the improvements in the overall sharpness and fine details.

