# OpenReview forum: "Pixel-Space Post-Training of Latent-Diffusion Models"
_ICLR.cc/2025/Conference — ICLR 2025 Conference Withdrawn Submission_

### Official Review · Reviewer_6kfJ · 2024-10-20

**Soundness:** 3
**Presentation:** 3
**Contribution:** 2
**Rating:** 6
**Confidence:** 2

**Summary:**

This paper think the flaws occured in latent diffusion models is resulted from all the training process are conducted in the low-resolution latent space. Thus this manuscript proposes to add pixel-space supervision in the post-training process. This idea is simply and general, so it can have both implementation in supervised finetuning and reward-based finetuning. In practice, the authors modify SFT and DPO algorithm by mapping the latent back to the pixel space, and compute the pixel-space SFT/DPO as an additional loss term. The experiments is abundant, including lots of human evaluation, and verifies its effectiveness.

**Strengths:**

1. This method have a relatively good motivation.

2. The writing is clean and easy to follow.

3. The overall idea is general, so it can be implemented in both SFT and reward based finetuning.

4. Although the method looks simple, extensive experiments show its effectiveness.

**Weaknesses:**

1. The used DiT model detail is not explained.

2. The method is quite simple. The analysis of why this method can improve performance is missed.

3.  The authors do not conduct their experiments on more recent model. It is suggested to verity their method in more recent UNet and DiT models.

Minor: The text in equation should be put inside \text{...}

**Questions:**

1. Which DiT model does the authors use?

2. Is this method very easy to implment, like change a minor architecture, and add a loss term in the original loss?

3. Why the authors conduct experiments in SD1.5 and an unknown DiT? Why they do not use more recent UNet like SDXL, and use DiT in SD3?

---

### Official Review · Reviewer_vZud · 2024-10-27

**Soundness:** 2
**Presentation:** 3
**Contribution:** 2
**Rating:** 3
**Confidence:** 4

**Summary:**

This paper addresses limitations in Latent Diffusion Models (LDMs) related to generating high-frequency details and complex compositions, which the authors attribute to conducting pre- and post-training exclusively within a low-resolution (8×8) latent space. To overcome these issues, the paper proposes integrating pixel-space supervision in the post-training phase, aiming to better capture high-frequency details.

**Strengths:**

1. Addressing the enhancement of detail realism in latent diffusion generation is an important problem.
2. Experimental results demonstrate that the proposed method achieves performance improvements over existing fine-tuning approaches.

**Weaknesses:**

1. The proposed technical approach—simply adding a pixel-level loss—lacks novelty and may appear overly simplistic. The use of reinforcement learning within a diffusion model is also not groundbreaking, and the extension of SIMPo does not constitute a notable contribution. It would be better if the authors could clarify why the pixel-level loss is novel.

2. Technical details and motivations are not well-articulated: (1) Why does Equation 2 use a decoded ground truth image rather than the original image? If decoding already introduces detail loss, it is unclear how this approach can effectively enhance detail realism as intended. (2) In applying the pixel loss, is a random timestep selected? If so, when the noise level is high, how is one-step prediction accuracy ensured? It would be better if the authors could clearly clarify these questions.


3. The results presented in Figure 4 reveal that the method still struggles with preserving finer details, particularly in high-frequency regions, where the generated images display notable artifacts or blurriness. This suggests that while the proposed pixel-level supervision may have improved some aspects of detail generation, it falls short in producing consistently realistic textures across different parts of the image. Additionally, certain areas of the images, which typically require precise detailing—such as edges, textures, or small intricate patterns—do not appear as natural as expected, raising questions about the method's overall effectiveness in enhancing detail realism.

4. The enhancement of natural image details is unconvincing. I suggest testing this approach in domains where detail accuracy is more critical, such as talking face generation. It would be beneficial to see if it truly improves fine details like eyes and teeth. The authors may consider using Diffusion-based foundation models, such as https://github.com/fudan-generative-vision/hallo.

**Questions:**

Please refer to the weaknesses.

---

### Official Review · Reviewer_1yCD · 2024-11-03

**Soundness:** 2
**Presentation:** 2
**Contribution:** 2
**Rating:** 3
**Confidence:** 3

**Summary:**

This paper hypothesizes that losses of details and artifacts in high-frequency details in Latent diffusion models (LDMs) are partially caused by training on the lower-resolution latent space, and proposes adding a pixel-space objective during LDMs post-training. The experiments show improvements in both DiT-based and UNet-based LDMs for reward-based and supervised fne-tuning.

**Strengths:**

1. The hypothesis that losses of details and artifacts in high-frequency details in Latent diffusion models (LDMs) are partially caused by training on the lower-resolution latent space is sound.
2. The proposed pixel-space objective is straightforward.

**Weaknesses:**

1. The proposed method, with additional supervision in the pixel space, is too simplistic and lacks technical innovation.
2. The proposed method lacks comprehensive experimental validation; current experiments rely solely on human evaluation without additional quantifiable metrics.
3. The proposed method has only been validated under fine-tuning settings, without verification under pre-training settings.
4. The proposed method is relatively costly, as transforming from the latent space to the pixel space involves passing through a VAE, which can be computationally intensive.
5. The proposed method lacks theoretical and experimental explanations for why it is effective.
6. The quality of the writing is suboptimal, exhibiting problems with both logical coherence and linguistic fluency, particularly evident in the introduction section.

**Questions:**

All my questions are in weaknesses above.

---

### Official Review · Reviewer_V1B9 · 2024-11-03

**Soundness:** 2
**Presentation:** 3
**Contribution:** 2
**Rating:** 5
**Confidence:** 5

**Summary:**

The paper proposes to adjust the supervised-finetuning originally proposed by Emu, by computing the loss in pixel space instead of latent space. Furthermore, the loss is on the reverse-noised sample (noise prediction removed from image), rather than on the noise prediction itself. They demonstrate the efficacy of this change via evaluation by third party human annotators.

**Strengths:**

[S1] The pixel-space SFT and pixel-space DPO consistently have superior image quality win rate to baselines and default SFT/DPO, along with similar or better text-alignment.

[S2] The paper communicates its key ideas thoroughly and clearly, and provides good examples and metrics that are sufficient to convince of the efficacy of the proposed approach.

[S3] The method is flexible and can easily be applied to any model.

**Weaknesses:**

[W1] The efficacy of the method is not very well-connected to the motivation. Pixel space is not necessary for such stunning or complex images (although it may increase the likelihood the model generates such images). Without zoom-in figures, it is unclear how/why the higher resolution loss is helpful, since it's not obvious that the fine-grained details are any better, which should be the main outcome. Simply put, I see no evidence that "adding pixel-space supervision in the post-training process to better preserve high-frequency details." (abstract)

[W2] Related to W1, there seems to be a significant confounding factor: instead of the only difference between SFT and yours being the latent vs. pixel space, Equations 1 and 2 indicate that SFT operates on noise predictions only, whereas your loss operates on the input with the noise prediction used for reverse diffusion. Thus, it is difficult to gauge the difference that should be attributed to computing loss in pixel space, compared to computing loss after removing the noise prediction.

[W3] The proposed method is a very incremental adjustment on SFT (the key insight from Emu). SFT is simply performed with 2 additional steps: subtracting the noise from the input, and decoding to pixel space for MSE in pixel space. Without other compelling insights, this is only very slightly novel.

[W4] While human measurement is undoubtedly the gold standard, it would have been helpful to provide some automatic metrics to contextualize the results.

**Questions:**

What is the difference in generated image quality when the loss is computed as in the paper (on actual images) versus on denoised latent? (where the denoising is done on the same schedule as for images)

What evidence is there that the high frequency details of themselves have changed, as opposed to the images just having superior layout/lighting/overall appeal?

---

### Note · Authors · 2024-11-14

I have read and agree with the venue's withdrawal policy on behalf of myself and my co-authors.